# (Unconstrained) Beam Search is Sensitive to Large Search Discrepancies

## Abstract

Beam search is the most popular inference algorithm for decoding neural sequence models. Unlike greedy search, beam search allows for a non-greedy local decisions that can potentially lead to a sequence with a higher overall probability. However, previous work found that the performance of beam search tends to degrade with large beam widths. In this work, we perform an empirical study of the behavior of the beam search algorithm across three sequence synthesis tasks. We find that increasing the beam width leads to sequences that are disproportionately based on early and highly non-greedy decisions. These sequences typically include a very low probability token that is followed by a sequence of tokens with higher (conditional) probability leading to an overall higher probability sequence. However, as beam width increases, such sequences are more likely to have a lower evaluation score. Based on our empirical analysis we propose to constrain the beam search from taking highly non-greedy decisions early in the search. We evaluate two methods to constrain the search and show that constrained beam search effectively eliminates the problem of beam search degradation and in some cases even leads to higher evaluation scores. Our results generalize and improve upon previous observations on copies and training set predictions.

## 1 Introduction

Neural sequence models are among the most popular tools for modeling sequential data and have been applied to a range of applications including machine translation (Gehring et al., 2017), summarization (Chopra et al., 2016), image captioning (Vinyals et al., 2017), and conversation modeling (Vinyals & Le, 2015). The most commonly used inference algorithm for decoding neural sequence models is beam search, a search algorithm that generates the sequence tokens one-by-one while keeping a fixed number of active candidates (beams) at each step.

Recently, several works reported the problem of performance degradation in beam search. In machine translation, Koehn & Knowles (2017) found that beam search "only improves translation for narrow beams and deteriorates when exposed to a larger search space". They chose this problem as one of six central challenges in machine translation. Ott et al. (2018) proposed the existence of training pairs in which the target is a copy of the source as an explanation for the performance degradation. For larger beams, more predictions can be classified as "copies"[1] and filtering these copies reduces the performance degradation.

In image captioning, Vinyals et al. (2017) observed performance degradation for wider beams and highlighted the use of a narrower beam search as one of the most significant improvements in their model. They hypothesized that the performance degradation is either due to overfitting or that the objective function used in training (likelihood) is not aligned with human judgement. Their analysis found that wider beams exhibited more predictions that repeat training captions and fewer novel ones. They claim that this observation supports the hypothesis that the model is overfitted and therefore see the use of smaller beam width as "another way to regularize".

In this work, we analyze the performance of beam search across multiple tasks including machine translation, abstractive summarization, and image captioning. We present an explanatory model that

---

[1] "Copies" are predictions that share at least 50% of their unigrams with their source (Ott et al., 2018).

is based on the concept of *search discrepancies* (deviations from greedy choices) and perform an empirical study of the distribution of such discrepancies. We make the following contributions:

1. We show that increasing the beam width leads to solutions with more and larger early discrepancies. These sequences often have lower evaluation score, leading to the observed performance degradation. As we increase the beam width, the difference between discrepancies that are associated with improved vs. degraded solutions grows substantially.

2. We show that our explanatory model generalizes the previously observed copies and predictions that repeat training set targets and accounts for more of the degraded predictions.

3. Exploiting the above insights, we propose a fix that is based on constraining the discrepancies considered by the beam search. An empirical analysis shows it successfully eliminates the performance degradation.

## 2 PRELIMINARIES

### 2.1 NEURAL SEQUENCE MODELS

Given a model parameterized by $\theta$ and an input $x$, the problem of sequence generation consists of finding a sequence $\hat{y}$ such that $\hat{y} = \arg\max_{y \in Y} P_\theta(y \mid x)$, where $Y$ is the set of all sequences. y is a sequence of tokens $y = \{y_0, ...y_{T-1}\}$ from vocabulary $\mathcal{V}$, where $T$ is the length of the sequence y. The expression $P_\theta(y \mid x)$ can then be factored as $P_\theta(y \mid x) = \prod_{t=0}^{T-1} P_\theta(y_t \mid x; \{y_0, ..., y_{t-1}\})$, or for convenience using log-probability as $\sum_{t=0}^{T-1} log P_\theta(y_t \mid x; \{y_0, ..., y_{t-1}\})$.

It is common to model $log P_\theta(y_t \mid x; \{y_0, ..., y_{t-1}\})$ using a Recurrent Neural Network (RNN), where the sequence $\{y_0, ..., y_{t-1}\}$ conditioned on is expressed by a fixed length hidden state $h_t$. This hidden state is updated using a non-linear function $f$: $h_{t+1} = f(h_t, y_t)$.

Exhaustive search to find the globally optimal sequence is not tractable. A greedy algorithm that selects the best candidate at each time step $y_t = \arg\max_{y \in \mathcal{V}} log P_\theta(y \mid x; \{y_0, ..., y_{t-1}\})$ makes a sequence of locally optimal decisions, but can lead to a globally sub-optimal sequence. Beam search, in contrast, extends the $B$ most probable partial solutions at each step, where $B$ is called *beam width*. Following Vijayakumar et al. (2018), we denote the set of $B$ solutions held by the beam search at step $t - 1$ as $Y_{[t-1]} = \{y_{1,[t-1]}, ..., y_{B,[t-1]}\}$. At each step, beam search selects the top scoring $B$ candidates from the set of all possible one token extensions of its beams $\mathcal{Y}_t = \{y_{[t]} \mid y_{[t-1]} \in Y_{[t-1]} \wedge y_t \in \mathcal{V}\}$. Formally, the beam search candidates are updated as follows:

$$Y_{[t]} = \arg\max_{y_{[1,t]}, ..., y_{[B,t]} \in \mathcal{Y}_t} \sum_{b \in [1..B]} log P_\theta(y_{b,t} \mid x) \tag{1}$$

$$s.t. \quad y_i \neq y_j \quad \forall i \neq j; \quad i, j \in [1..B]$$

### 2.2 SEARCH DISCREPANCIES IN NEURAL SEQUENCE GENERATION

In combinatorial search, a search discrepancy is a decision made by the search algorithm that is not the most highly rated one according to the heuristic (Harvey & Ginsberg, 1995). In the context of search for neural sequence generation, we define a search discrepancy as extending a partial sequence with a token that is not the most probable one (i.e., different than the greedy algorithm). More formally, a sequence y is considered to have a search discrepancy at time step $t$ if

$$log P_\theta(y_t \mid x; \{y_0, ..., y_{t-1}\}) < \max_{y \in \mathcal{V}} log P_\theta(y \mid x; \{y_0, ..., y_{t-1}\}). \tag{2}$$

We denote the ratio between the most probable token and the chosen token as *discrepancy gap*. We measure the gap based on the difference in log-probability, i.e., the discrepancy gap at step $t$ is

$$\max_{y \in \mathcal{V}} log P_\theta(y \mid x; \{y_0, ..., y_{t-1}\}) - log P_\theta(y_t \mid x; \{y_0, ..., y_{t-1}\}). \tag{3}$$

## 3 EXPERIMENTAL SETUP

We perform an extensive empirical evaluation over multiple tasks, models, datasets, toolkits, and evaluation metrics. Following is a description of the experimental setup for each task.

**Machine Translation.** We use the convolutional model by Gehring et al. (2017) implemented in the *fairseq-py* toolkit. We present results for two models, trained on WMT'14 En-Fr and En-De datasets and evaluated on newstest2014 En-Fr and En-De, respectively.

**Summarization.** We use the abstractive summarization model by Chopra et al. (2016) implemented in OpenNMT toolkit (Klein et al., 2017). The model is trained and evaluated using Rush et al.'s (2015) test split of the Gigaword corpus (Graff et al., 2003).

**Image Captioning.** We use the model by Vinyals et al. (2017), trained on the MSCOCO dataset (Lin et al., 2014). The test set includes 5000 images based on Karpathy & Fei-Fei's (2015) splits.

In machine translation and summarization, we apply length normalization on the hypotheses log-likelihood, as it was shown to reduce the performance degradation by not prioritizing short sentences (Koehn & Knowles, 2017; Gehring et al., 2017). For image captioning, consistent with previous works, we do not use length normalization (we also found it reduces the overall performance).

## 3.1 EVALUATION METRICS

While beam search finds the (approximately) most probable sequence, the quality of a sequence is evaluated based on human references using a task-specific evaluation metric. For machine translation and image captioning we use BLEU-$n$ (Papineni et al., 2002), a geometric average of precision over 1- to $n$-grams multiplied by a brevity penalty for short sentences. As in recent literature, we present results for BLEU-4. Corpus-level BLEU is reported without smoothing, while for sentence-level BLEU we use smoothed $n$-gram counts for $n > 1$ (Lin & Och, 2004). For image captioning, we also evaluated the performance using CIDEr (Vedantam et al., 2015) and SPICE (Anderson et al., 2016b) and report these metrics in Appendix E.

For summarization, we use ROUGE (Lin, 2004), the $n$-gram recall between candidate summary and a reference. We report the F-score of ROUGE-1, however similar trends were observed for the F-score of ROUGE-L (for longest common subsequence).

## 4 EMPIRICAL ANALYSIS OF SEARCH DISCREPANCIES IN BEAM SEARCH

In this section we present an empirical analysis of the search discrepancies. We analyze and compare the most likely hypotheses found by a beam search for the following beam widths: {1, 3, 5, 25, 100, 250}. Due to space, we present detailed results for one of the tasks and summarize the results for the others. The results for all tasks and metrics can be found in Appendix A.

## 4.1 BASELINE RESULTS

Table 1 presents the performance of beam search with different beam widths, based on the chosen evaluation metrics. The performance degradation for larger beam widths appears for all tested tasks based on their task-specific evaluation metric. These results are consistent with the existing reports of such performance degradation (Koehn & Knowles, 2017; Ott et al., 2018; Vinyals et al., 2017).

Table 1: Baseline results for different beam widths (higher values are better, best results in bold).

| Task | Dataset | Metric | $B=1$ | $B=3$ | $B=5$ | $B=25$ | $B=100$ | $B=250$ |
|------|---------|--------|-------|-------|-------|--------|---------|---------|
| Translation | En-De | BLEU4 | 25.27 | 26.00 | **26.11** | 25.11 | 23.09 | 21.38 |
| | En-Fr | BLEU4 | 40.15 | 40.77 | **40.83** | 40.52 | 38.64 | 35.03 |
| Summarization | Gigaword | R-1 F | 33.56 | **34.22** | 34.16 | 34.01 | 33.67 | 33.23 |
| Captioning | MSCOCO | BLEU4 | 29.66 | **32.36** | 31.96 | 30.04 | 29.87 | 29.79 |

## 4.2 THE DISTRIBUTION OF SEARCH DISCREPANCIES

In this section, we analyze the distribution and size of search discrepancies vs. their position (index) in sequence. Figure 1 shows the number of discrepancies per position for the most likely hypotheses generated by a beam search on the WMT'14 En-De test set for different beam widths (all graphs

are based on the same number of solutions, however the total number of discrepancies in the generated solutions is not necessarily the same for different beam widths). In general, the majority of discrepancies happen in early positions. More interestingly, for larger beams, the number of early discrepancies grows significantly while the number of later discrepancies stays approximately the same. Larger beams seem to allow the search to find solutions with higher overall probability by exploring less probable early tokens, however they do not seem to lead to more probable sequences that share a prefix with solutions found for a smaller beam width. Similar results for the other tasks are reported in Appendix A. For image captioning (MSCOCO), we find the majority of early search discrepancies appear on the second token due to the first token being "a" with high probability in almost all sentences (in greedy search, for example, 99% of the generated captions start with "a").

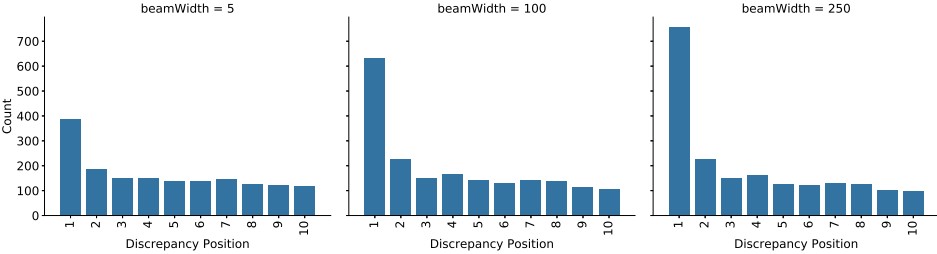

Figure 1: WMT'14 En-De: Distribution of discrepancy positions for different beam widths.

Next, we analyze the discrepancy gap vs. sequence position. Figure 2 presents the mean gap per position for WMT'14 En-De for different beam widths. Again, we can see that the changes are mainly in the early positions: as we increase the beam width, the search tends to find solutions with larger early discrepancy gap, i.e., the early tokens are relatively less likely. The discrepancy gap of the other tokens remains similar. Similar results for the other tasks are reported in Appendix A.

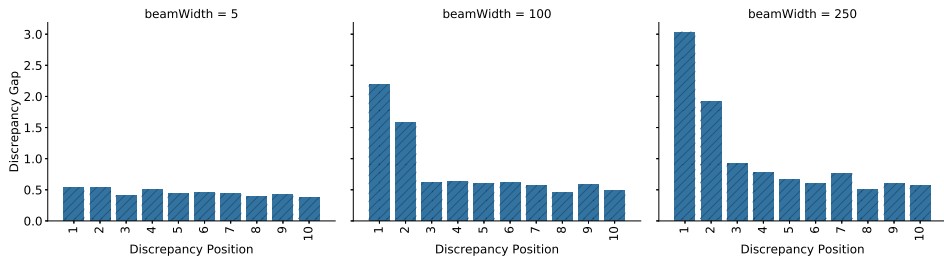

Figure 2: WMT'14 En-De: Mean discrepancy gap per position for different beam widths.

The increase in count and size of early discrepancies for larger beams means that the search manages to find solutions with higher overall probability when starting from a large discrepancy. However, these solution are not necessarily better according to the evaluation metric. The observed performance degradation suggests that the more probable solutions found by larger beams are, in fact, worse. Identifying discrepancies that are likely to lead to a worse solution is therefore a key task in addressing the performance degradation. In the next section, we analyze the differences between discrepancies in solutions with higher evaluation vs. solutions with lower evaluation.

### 4.3 IMPROVED VS. DEGRADED SOLUTIONS

We now compare the solutions generated by a greedy search with the solutions generated by beam search with different widths. We then analyze the discrepancies in solutions that were improved by increasing the beam width (with respect to the evaluation metric) vs. solutions that were degraded.

Figure 3 shows the number of discrepancies per position for WMT'14 En-De, comparing solutions that were improved vs. solutions that were degraded. For $B=5$ there are 386 solutions in which the first token is not based on a greedy decision. Of those, 200 have a better evaluation than the greedy solution and 169 have a worse evaluation. However, as we increase the beam width, we see that the increase in early discrepancies observed in Figure 1 is associated almost entirely with degraded

solutions. This result explains the observed performance degradation for larger beam widths. Similar results for the other tasks are reported in Appendix A.

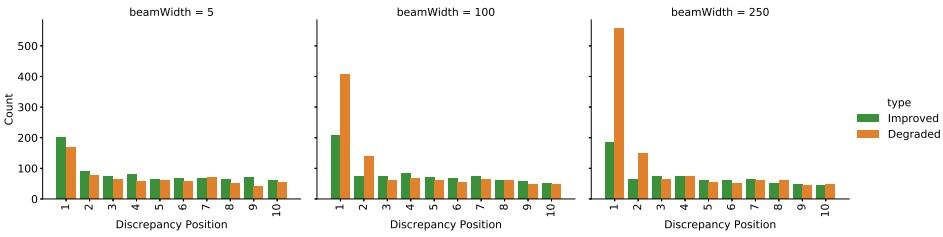

Figure 3: WMT'14 En-De: Distribution of discrepancy positions for different beam widths.

Next, we compare the discrepancy gaps in degraded vs. improved solutions. Figure 4 presents the mean discrepancy gap per position for the WMT'14 En-De dataset, for both the improved and the degraded solutions. Interestingly, we find that the additional early discrepancies that are associated with degraded solutions tend to have a much higher discrepancy gap compared to the ones associated with improved solutions. Similar results for the other tasks are reported in Appendix A.

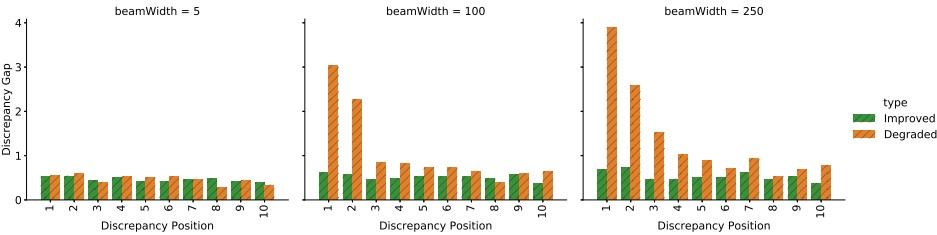

Figure 4: WMT'14 En-De: Mean discrepancy gap per position for different beam widths.

### 4.4 SEARCH DISCREPANCIES AND THE MOST LIKELY HYPOTHESIS

In order for a sequence with early large discrepancy to be selected by a beam search as (approximately) the most likely hypothesis, it has to be followed by tokens with higher (conditional) probability. Figure 5 shows the average (conditional) token probability for WMT'14 En-De (we use log-scale on the x axis to highlight the early positions). For larger beam widths, the average probability of early tokens decreases (due to larger discrepancy gaps) while the average probability of later tokens increases explaining the overall higher probability.[2] Figure 5 also shows the same graph for the improved vs. degraded solutions (compared to greedy search). For improved solutions, we do not see significant change as we increase the beam width. For degraded solutions, however, as we increase the beam width we find more and more early discrepancies that lead to an overall higher probability but a worse evaluation metric value. For all tasks, we found the differences for the degraded solutions to be larger than the improved solutions (see Appendix A).

Ott et al. (2018) observed the same pattern for copies, i.e., they have low first token probability and higher probabilities for subsequent tokens. Our analysis accounts for this behavior and suggests that copies are one instance of a more general pattern that leads to degraded sequences. In the next section, we show that our analysis generalizes copies, as well as training set predictions, and even accounts for additional degraded sequences.

### 4.5 GENERALIZING COPIES AND TRAINING SET PREDICTIONS

Table 2 shows the number of copies in machine translation and training set predictions in summarization and image captioning. For larger beams, the number of copies and training set predictions grows. Table 2 also reports the mean discrepancy gap of the first token (second token for MSCOCO, see Section 4.2). As our analysis predicts, the early gap of these predictions also grows significantly.

---

[2]When length normalization is not used, we should compare the product of token probabilities rather than the average token probabilities. See Appendix A.3 for results on the unnormalized image captioning task.

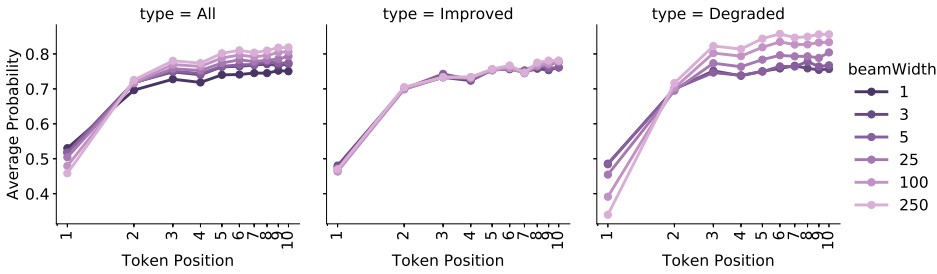

Figure 5: Average token probability per position for different beam widths.

Table 2: Number of copies and training set examples and the average first token discrepancy gap.

|  |  | $B{=}1$ | $B{=}3$ | $B{=}5$ | $B{=}25$ | $B{=}100$ | $B{=}250$ |
|---|---|---|---|---|---|---|---|
| En-De | # Copies | 23 | 40 | 49 | 179 | 385 | 567 |
| En-De | First token gap (copies) | 0.0 | 0.12 | 0.28 | 1.79 | 3.05 | 3.71 |
| En-De | First token gap (all) | 0.0 | 0.05 | 0.07 | 0.18 | 0.46 | 0.77 |
| En-Fr | # Copies | 25 | 28 | 41 | 89 | 227 | 358 |
| En-Fr | First token gap (copies) | 0.0 | 0.12 | 0.31 | 1.69 | 3.68 | 4.38 |
| En-Fr | First token gap (all) | 0.0 | 0.04 | 0.05 | 0.10 | 0.32 | 0.60 |
| Gigaword | # Training set predictions | 81 | 86 | 86 | 115 | 163 | 224 |
| Gigaword | First token gap (train pred.) | 0.0 | 0.07 | 0.07 | 0.98 | 1.84 | 2.61 |
| Gigaword | First token gap (all) | 0.0 | 0.12 | 0.12 | 0.29 | 0.39 | 0.55 |
| MSCOCO | # Training set predictions | 163 | 260 | 371 | 588 | 582 | 576 |
| MSCOCO | Second token gap (train pred.) | 0.0 | 0.39 | 0.87 | 1.76 | 1.82 | 1.82 |
| MSCOCO | Second token gap (all) | 0.0 | 0.20 | 0.29 | 0.49 | 0.51 | 0.51 |

Note that copies and training set predictions only partially account for the beam performance degradation. In WMT'14 En-De machine translation with $B = 25$, we find that copies account for $\approx 40\%$ of degraded solutions with first token gap. In Gigaword summarization with similar beam width, we find that training set examples account for $\approx 68\%$ of degraded solutions with first token gap. Furthermore, in MSCOCO, since many of the solutions in the greedy search, as well as many of the improved sequences, are training set captions, eliminating them all together is not desired. Instead, we are interested in avoiding the training set captions in the larger beam widths that led to the performance degradation. These, as Table 2 shows, have larger difference in the discrepancy gap.

## 4.6 An Illustrative Example

Consider the following example of training set predictions in Gigaword summarization. As we increase the beam width, we find more predictions with the structure: "⟨weekday⟩'s sports scoreboard" (Table 3).[3] As expected, these predictions have a large early discrepancy, followed by highly (conditionally) probable tokens. For $B = 100$, the average first token discrepancy gap for these summaries is $\approx 3.63$ compared to $\approx 0.39$ in the full test set. As none of the test references includes "sports scoreboard", these summaries have low evaluation score.

Table 3: Frequency of predicted summary for different beam widths.

|  | $B = 1$ | $B = 3$ | $B = 5$ | $B = 25$ | $B = 100$ | $B = 250$ |
|---|---|---|---|---|---|---|
| "⟨weekday⟩'s sports scoreboard" | 0 | 0 | 1 | 17 | 19 | 19 |

As a potential explanation for this phenomenon, we find that all texts that were summarized as "⟨weekday⟩'s sports scoreboard" included the corresponding weekday. In the training set, we found

---

[3]Without length normalization, the numbers are higher as this sequence is shorter than most summaries.

that in 2962 of the 2971 texts that were summarized to "⟨weekday⟩'s sports scoreboard" included the corresponding weekday. This can lead to the weekday's token suggested as a first token with a low, but sufficiently high, probability to get into the top $B$ tokens. Followed by high probability tokens, it can, in some cases, have an overall probability that is higher than the alternatives.

## 5 DISCREPANCY-CONSTRAINED BEAM SEARCH

Building on our empirical analysis, we propose to constrain the beam search to avoid considering highly unlikely discrepancies. To do so, we evaluate two methods of constraining the beam search:

1. **Discrepancy gap:** Given a threshold $\mathcal{M}$, we modify beam search to only consider candidates with a discrepancy gap smaller or equal to $\mathcal{M}$. Formally, we modify Eq. 1 to include the constraint $\max_{y \in \mathcal{V}} \log P_\theta(y \mid x; \{y_0, ..., y_{t-1}\}) - \log P_\theta(y_t \mid x; \{y_0, ..., y_{t-1}\}) \leq \mathcal{M}$.

2. **Beam candidate rank:** Given a threshold $\mathcal{N}$, we modify $\mathcal{Y}_t$ to only include the top $\mathcal{N}$ one token extensions in each beam. Note that the beam search still retains the top $B$ candidates, however it will not consider more than $\mathcal{N}$ candidates from the same beam.

Using the setup in Section 4.1, we compare these methods to the baseline. Although the analysis in Section 4 was done on the test set (to account for the performance degradation that was previously observed on the test set), $\mathcal{M}$ and $\mathcal{N}$ are tuned on a held-out validation set and no information from the test set was used to tune our methods.

As shown in Table 4, both methods significantly reduce, and in some cases completely eliminate, the performance degradation. In translation and summarization, we improve performance compared to baseline with the best test beam width. In general, the gap constraint seems to be perform better (most notably, for MSCOCO). The gap constraint allows for a finer-grained control over the accepted discrepancies, however the rank constraint is simpler and easier to tune.

Table 4: A comparison of the baseline results vs. the constrained beam search methods (higher values are better, best baseline results in bold).

| Dataset | Method | Threshold | $B$=1 | $B$=3 | $B$=5 | $B$=25 | $B$=100 | $B$=250 |
|---|---|---|---|---|---|---|---|---|
| En-De (BLEU-4) | Baseline | | 25.27 | 26.00 | **26.11** | 25.11 | 23.09 | 21.38 |
| | Constr. Gap | $\mathcal{M} = 1.5$ | 25.27 | 26.00 | 26.18 | 26.18 | 26.22 | 26.29 |
| | Constr. Rank | $\mathcal{N} = 2$ | 25.27 | 26.07 | 26.01 | 26.08 | 26.10 | 26.10 |
| En-Fr (BLEU-4) | Baseline | | 40.15 | 40.77 | **40.83** | 40.52 | 38.64 | 35.03 |
| | Constr. Gap | $\mathcal{M} = 2.0$ | 40.15 | 40.78 | 40.86 | 40.98 | 41.05 | 41.06 |
| | Constr. Rank | $\mathcal{N} = 3$ | 40.15 | 40.77 | 40.81 | 40.99 | 41.05 | 41.02 |
| Gigaword (R-1 F) | Baseline | | 33.56 | **34.22** | 34.16 | 34.01 | 33.67 | 33.23 |
| | Constr. Gap | $\mathcal{M} = 0.85$ | 33.56 | 34.27 | 34.29 | 34.43 | 34.33 | 34.32 |
| | Constr. Rank | $\mathcal{N} = 2$ | 33.56 | 34.48 | 34.45 | 34.25 | 34.23 | 34.32 |
| MSCOCO (BLEU-4) | Baseline | | 29.66 | **32.36** | 31.96 | 30.04 | 29.87 | 29.79 |
| | Constr. Gap | $\mathcal{M} = 0.45$ | 29.66 | 32.24 | 32.33 | 32.36 | 32.35 | 32.35 |
| | Constr. Rank | $\mathcal{N} = 2$ | 29.66 | 32.52 | 31.97 | 30.88 | 30.87 | 30.87 |

We also compared the number of copies and training set predictions in the baseline vs. the two discrepancy-constrained variants of beam search. We find that the constrained methods reduce the growth in the number of copies and training set predictions that happen as we increase the beam width. The detailed comparison can be found in Appendix B.

Finally, we repeated our analysis above and find that both constrained beam search variations substantially reduce the discrepancy phenomena observed in Section 4. Complete results and graphs for both constrained methods on WMT'14 En-De are in Appendix C (other tasks exhibited similar results).

## 6 DISCUSSION

Our results show that larger beam width leads to increasingly large early discrepancies. These very unlikely early tokens are later compensated by subsequent tokens with a much higher (conditional) probability compared to the subsequent tokens of the more probable early tokens. The large difference in the conditional probability of the subsequent tokens is at the heart of the observed performance degradation. Previous work has highlighted two potential biases that can account for this difference. *Exposure bias* (Ranzato et al., 2015) occurs since the model is only exposed to the training data and can be biased towards the training set distribution (our illustrative example demonstrates such bias due to a repetitive pattern in the training data). *Label bias* (Wiseman & Rush, 2016) occurs since token probabilities at each time step are locally normalized and therefore the successors of incorrect histories receive the same probability mass as the successors of a correct history.

These above biases help explain the observed behavior with large beam width: a biased (conditional) probability that concentrates high probability mass on one token and is locally normalized to sum to one compensates for earlier low probability tokens. The negative effects of these biases have been discussed before, however the described connection to the performance degradation in beam search and the explanatory framework to allow such analysis is, to our best knowledge, novel.

The use of the search discrepancy concept from heuristic and combinatorial search views the probabilities predicted by the neural network as a heuristic to guide the search. Early mistakes in such search have been shown to have a large negative effect on performance (Gent & Walsh, 1994). Substantial work has analyzed and proposed techniques to mitigate the phenomenon (Gomes et al., 2005; Cohen & Beck, 2018), including limited discrepancy search (Harvey & Ginsberg, 1995). Further investigation of the connection between such work and neural decoding may lead to further insight.

## 7 RELATED WORK

Search discrepancies have been the base of many search techniques in combinatorial search and optimization (e.g., Harvey & Ginsberg, 1995; Walsh, 1997; Beck & Perron, 2000). Furcy & Koenig (2005) proposed BULB, a complete variant of beam search that backtracks based on search discrepancies, as a memory-efficient alternative to best-first heuristic search for path-finding problems.

Several works have modified or constrained beam search for different purposes. Vijayakumar et al. (2018) changed the objective to allow diverse decoding. Hokamp & Liu (2017) proposed grid beam search to support lexical constraints. Anderson et al. (2016a) proposed a constrained beam search that forces inclusion of selected tokens in the output. Freitag & Al-Onaizan (2017) analyzed pruning techniques for beam search in machine translation. Their strategy of limiting "maximum candidates per node" is similar to the rank constraint in our work, however their analysis is focused on speeding up beam search rather than addressing the phenomenon of performance degradation.

A recent line of work in machine translation suggested the performance degradation is due to length bias (Yang et al., 2018; Murray & Chiang, 2018). For larger beams, an end-of-sentence token with a lower probability that leads to an overall more probable hypothesis is more likely to be considered by the beam search. However, we showed peformance degradation above even when using length normalization and in tasks where length bias does not appear (see Appendix D for more details).

## 8 CONCLUSION

In this work, we perform an empirical analysis of the performance degradation in beam search across three neural sequence decoding tasks. We find that the performance degradation for large beam widths is due to the increasing number of early and large search discrepancies. Our analysis generalizes previous results including the existence of copy predictions in machine translation and the training set predictions in image captioning, and accounts for additional degraded sequences. Based on our analysis, we propose two constrained variants of beam search that avoid large discrepancy gaps and successfully eliminate the performance degradation in beam search.

This work provides a deeper understanding of beam search behavior and an analytical framework to study search discrepancies. We believe the analysis, the framework, and the proposed solutions can support the development of better training methods and decoding algorithms.

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

# A    FULL EMPIRICAL ANALYSIS

## A.1    MACHINE TRANSLATION ON WMT'14 EN-FR

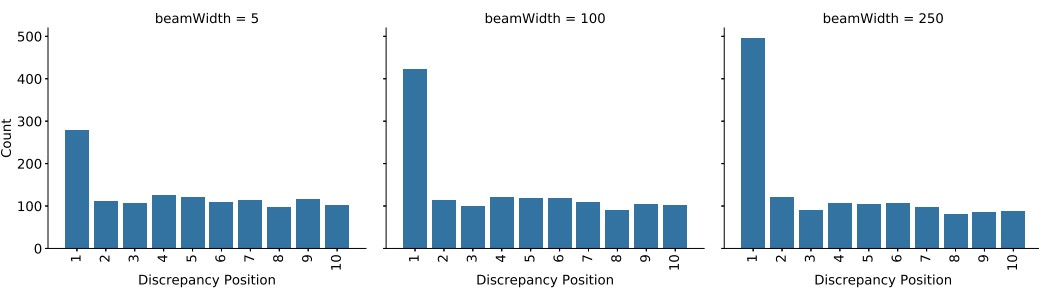

Figure 6: WMT'14 En-Fr: Distribution of discrepancy positions for different beam widths.

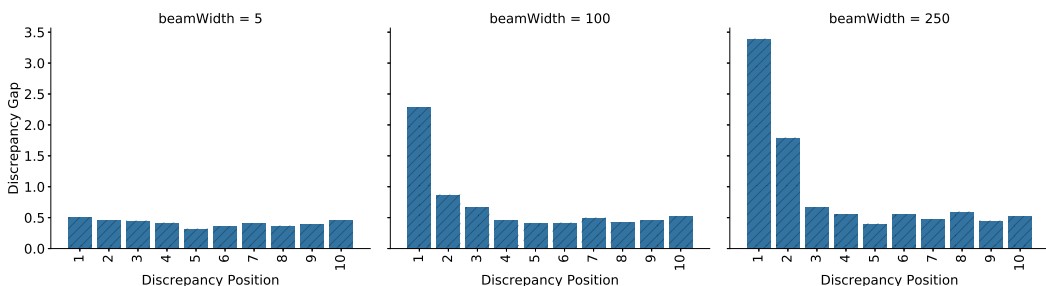

Figure 7: WMT'14 En-Fr: Mean discrepancy gap per position for different beam widths.

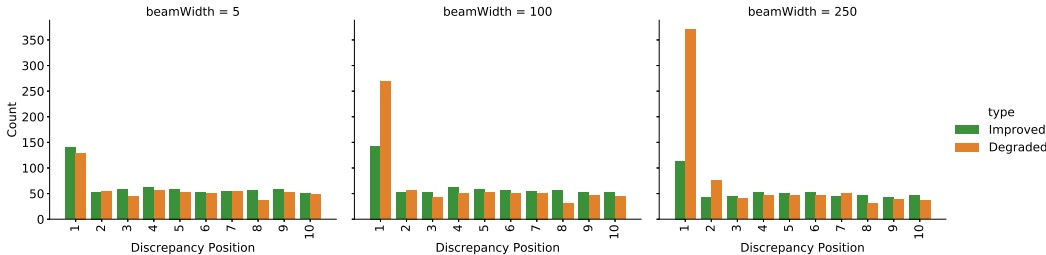

Figure 8: WMT'14 En-Fr: Distribution of discrepancy positions for different beam widths.

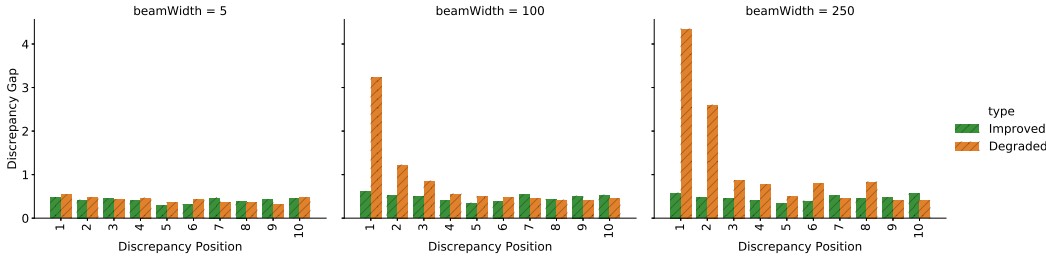

Figure 9: WMT'14 En-Fr: Mean discrepancy gap per position for different beam widths.

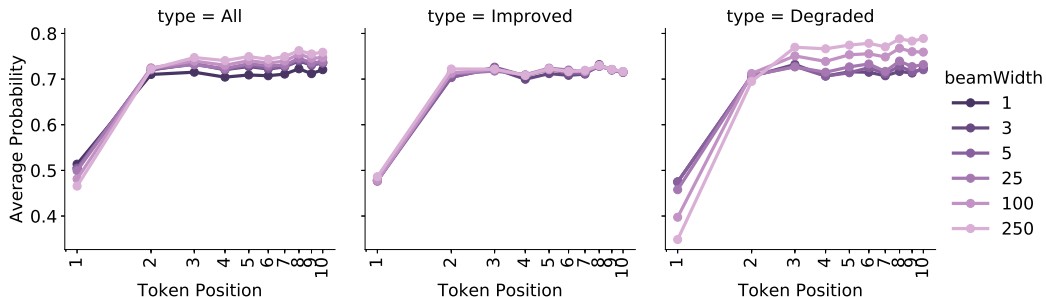

Figure 10: WMT'14 En-Fr: Average token probability per position for different beam widths.

## A.2 SUMMARIZATION ON GIGAWORD

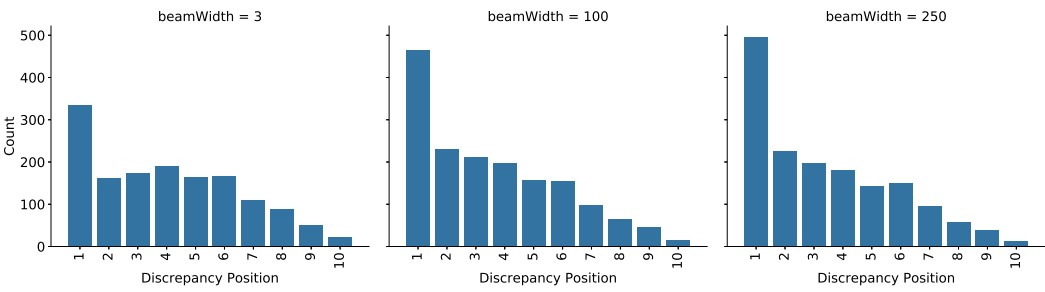

Figure 11: Gigaword: Distribution of discrepancy positions for different beam widths.

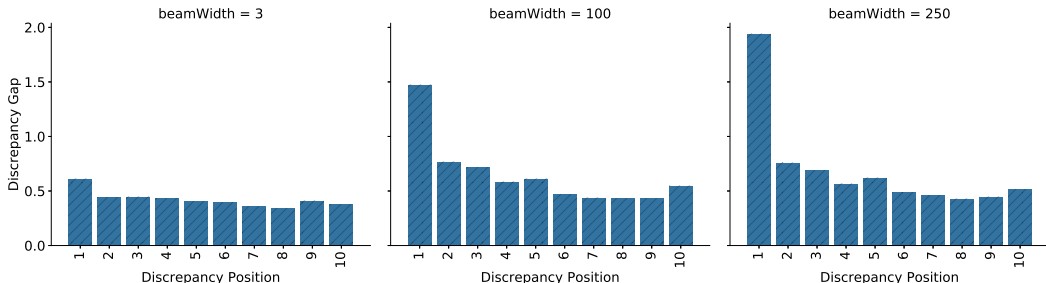

Figure 12: Gigaword: Mean discrepancy gap per position for different beam widths.

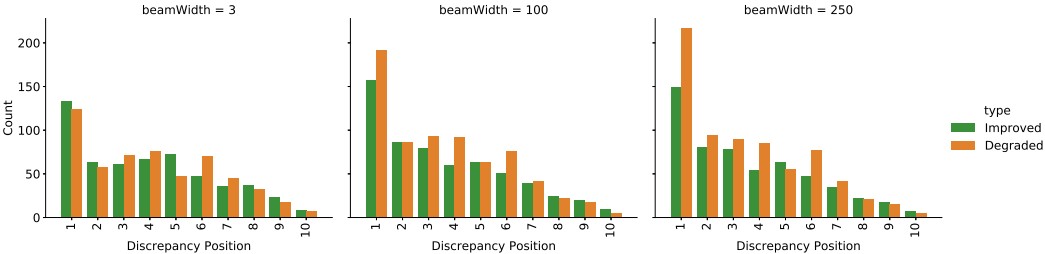

Figure 13: Gigaword: Distribution of discrepancy positions for different beam widths.

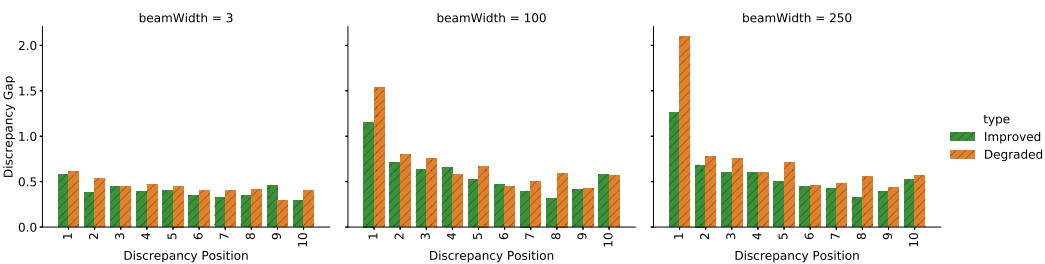

Figure 14: Gigaword: Mean discrepancy gap per position for different beam widths.

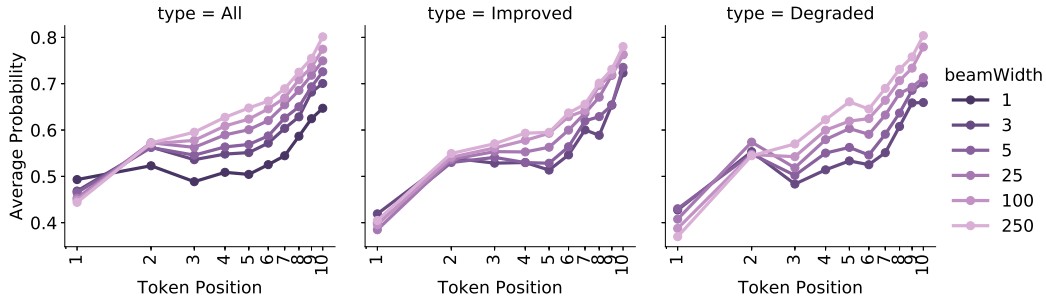

Figure 15: Gigaword: Average token probability per position for different beam widths.

## A.3 IMAGE CAPTIONING ON MSCOCO

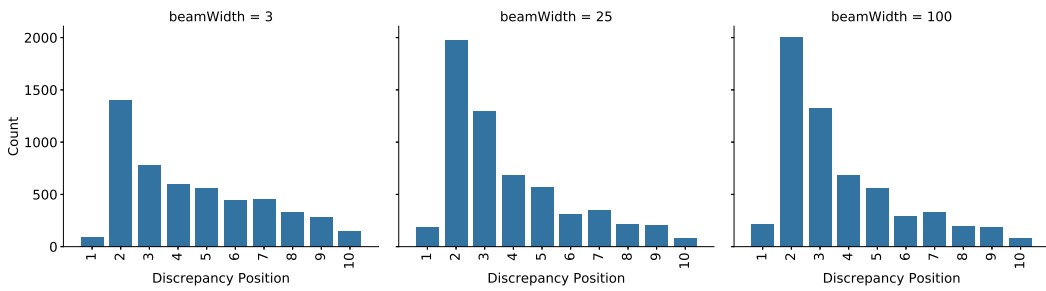

Figure 16: MSCOCO: Distribution of discrepancy positions for different beam widths.

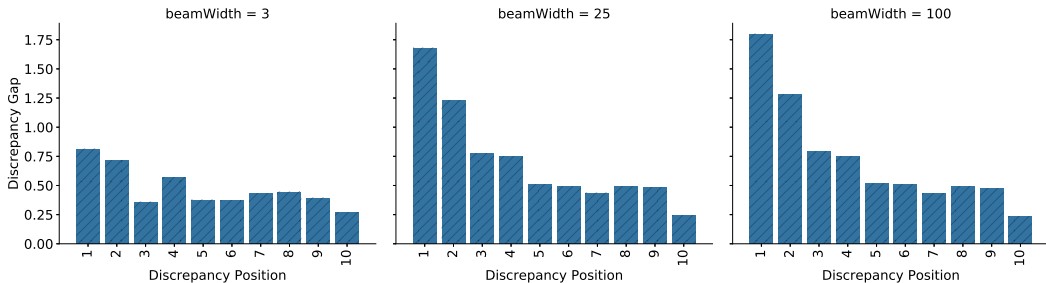

Figure 17: MSCOCO: Mean discrepancy gap per position for different beam widths.

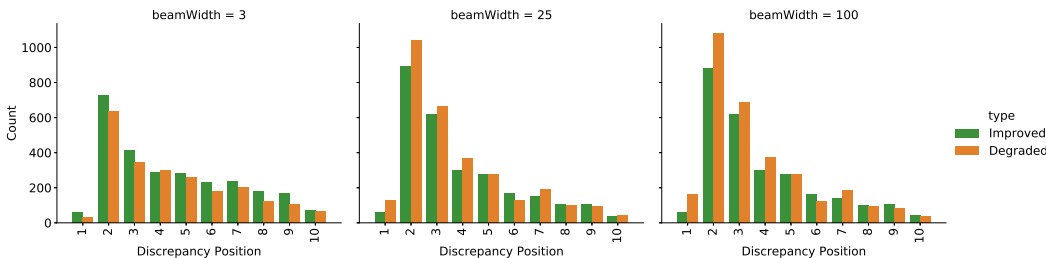

Figure 18: MSCOCO: Distribution of discrepancy positions for different beam widths.

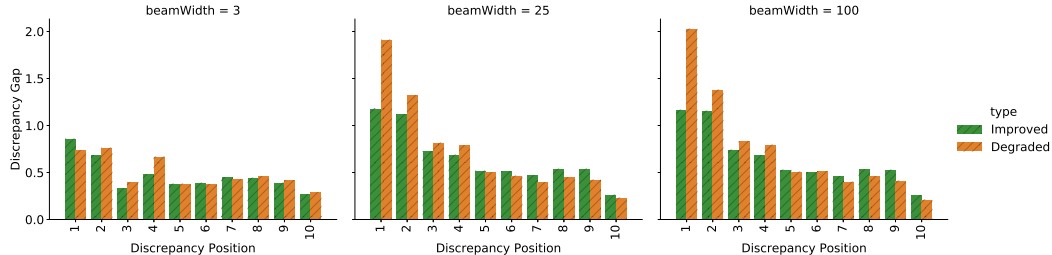

Figure 19: MSCOCO: Mean discrepancy gap per position for different beam widths.

Table 5: MSCOCO: Probability of the early (first two) tokens vs. the probability of the rest.

| Beam | All | | Improved | | Degraded | |
|---|---|---|---|---|---|---|
| | Early | Rest | Early | Rest | Early | Rest |
| $B{=}1$ | -1.48 | -6.98 | N/A | N/A | N/A | N/A |
| $B{=}3$ | -1.68 | -5.11 | -1.76 | -5.32 | -1.78 | -5.09 |
| $B{=}25$ | -2.02 | -4.09 | -2.04 | -4.26 | -2.21 | -3.99 |
| $B{=}100$ | -2.07 | -4.02 | -2.06 | -4.21 | -2.30 | -3.91 |
| $B{=}250$ | -2.08 | -4.01 | -2.06 | -4.20 | -2.31 | -3.90 |

# B  COPIES AND TRAINING SET PREDICTIONS IN DISCREPANCY-CONSTRAINED BEAM SEARCH

Table 6 compares the number of copies in the baseline vs. the discrepancy-constrained methods for the machine translation tasks for each beam width. For the baseline, we can see that as we increase the beam width, the number of copies grows significantly. However, both discrepancy-constrained methods significantly reduce this growth.

Table 6: Number of copies in machine translations for the baseline and the two types of discrepancy-constrained beam search for different beam widths.

| Dataset | Method | Parameter | $B{=}1$ | $B{=}3$ | $B{=}5$ | $B{=}25$ | $B{=}100$ | $B{=}250$ |
|---|---|---|---|---|---|---|---|---|
| En-De | Baseline | | 23 | 40 | 49 | 179 | 385 | 567 |
| En-De | Constr. Gap | $\mathcal{M} = 1.5$ | 23 | 39 | 42 | 50 | 53 | 55 |
| En-De | Constr. Rank | $\mathcal{N} = 2$ | 23 | 38 | 44 | 46 | 54 | 55 |
| En-Fr | Baseline | | 25 | 28 | 41 | 89 | 227 | 358 |
| En-Fr | Constr. Gap | $\mathcal{M} = 2.0$ | 25 | 27 | 37 | 43 | 46 | 45 |
| En-Fr | Constr. Rank | $\mathcal{N} = 3$ | 25 | 28 | 38 | 42 | 42 | 46 |

Table 7 compares the number of training set predictions in the baseline vs. the discrepancy-constrained methods for the summarization and image captioning tasks for each beam width. For

the baseline, we can see that as we increase the beam width, the number of training set predictions grows significantly. However, as with copies, both discrepancy-constrained methods significantly reduce the growth in training set predictions.

Table 7: Number of predictions that are in the training set for the baseline and the two types of discrepancy-constrained beam search for different beam widths.

| Dataset | Method | Parameter | $B=1$ | $B=3$ | $B=5$ | $B=25$ | $B=100$ | $B=250$ |
|---------|--------|-----------|-------|-------|-------|--------|---------|---------|
| Gigaword | Baseline | | 81 | 86 | 86 | 115 | 163 | 224 |
| Gigaword | Constr. Gap | $\mathcal{M}=0.85$ | 81 | 81 | 77 | 79 | 78 | 78 |
| Gigaword | Constr. Rank | $\mathcal{N}=2$ | 81 | 81 | 79 | 79 | 79 | 79 |
| MSCOCO | Baseline | | 163 | 260 | 371 | 588 | 582 | 576 |
| MSCOCO | Constr. Gap | $\mathcal{M}=0.45$ | 163 | 265 | 271 | 271 | 271 | 271 |
| MSCOCO | Constr. Rank | $\mathcal{N}=2$ | 163 | 242 | 262 | 231 | 231 | 231 |

## C RESULTS FOR CONSTRAINED BEAM SEARCH ON WMT'14 EN-DE

### C.1 RESULTS FOR DISCREPANCY GAP CONSTRAINED BEAM SEARCH ($\mathcal{M}=1.5$)

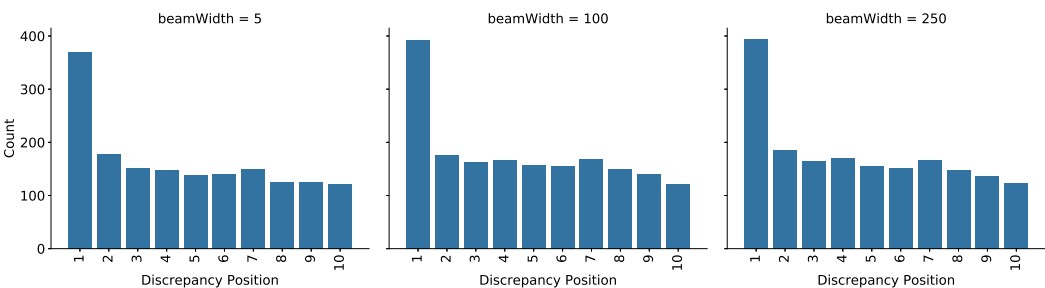

Figure 20: WMT'14 En-De: Distribution of discrepancy positions ($\mathcal{M}=1.5$).

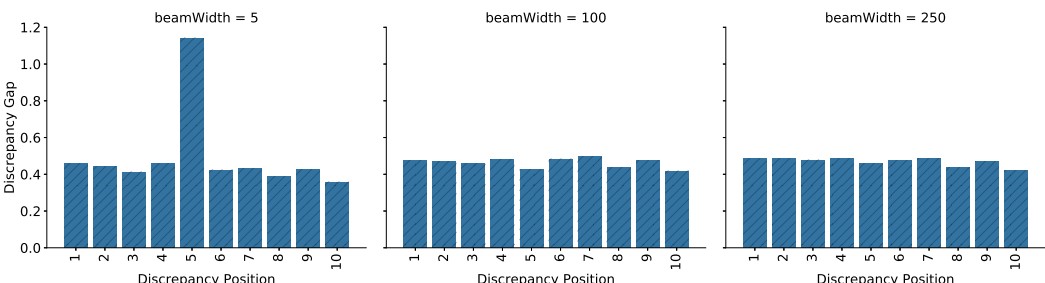

Figure 21: WMT'14 En-De: Mean discrepancy gap per position ($\mathcal{M}=1.5$).

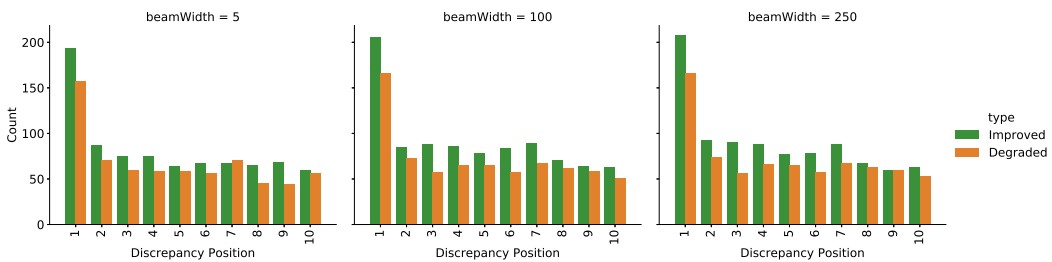

Figure 22: WMT'14 En-De: Distribution of discrepancy positions ($\mathcal{M} = 1.5$).

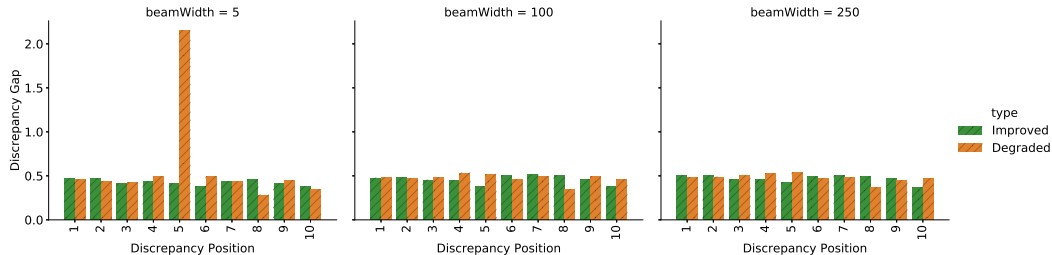

Figure 23: WMT'14 En-De: Mean discrepancy gap per position ($\mathcal{M} = 1.5$).

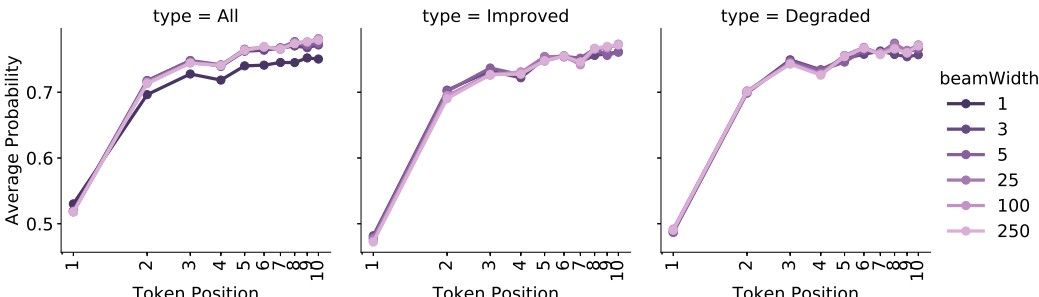

Figure 24: WMT'14 En-De: Average token probability per position ($\mathcal{M} = 1.5$).

## C.2 RESULTS FOR RANK CONSTRAINED BEAM SEARCH ($\mathcal{N} = 2$)

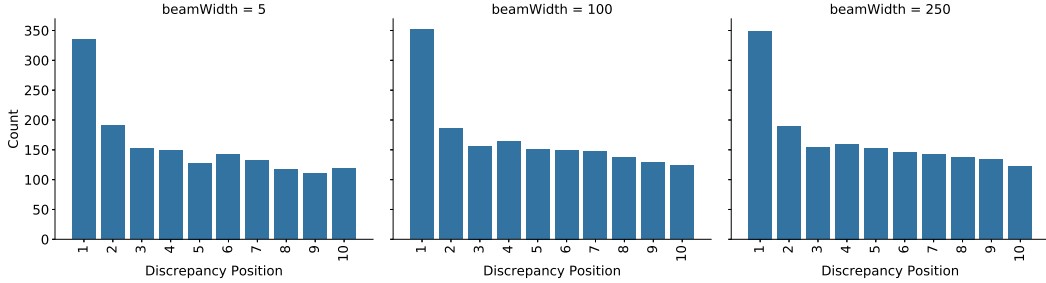

Figure 25: WMT'14 En-De: Distribution of discrepancy positions ($\mathcal{N} = 2$).

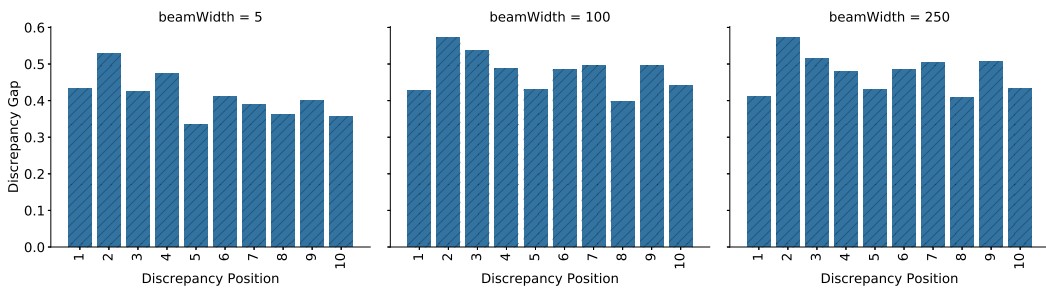

Figure 26: WMT'14 En-De: Mean discrepancy gap per position ($\mathcal{N} = 2$).

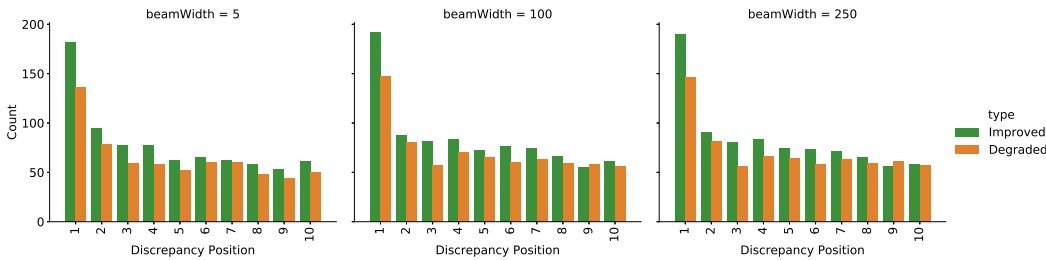

Figure 27: WMT'14 En-De: Distribution of discrepancy positions ($\mathcal{N} = 2$).

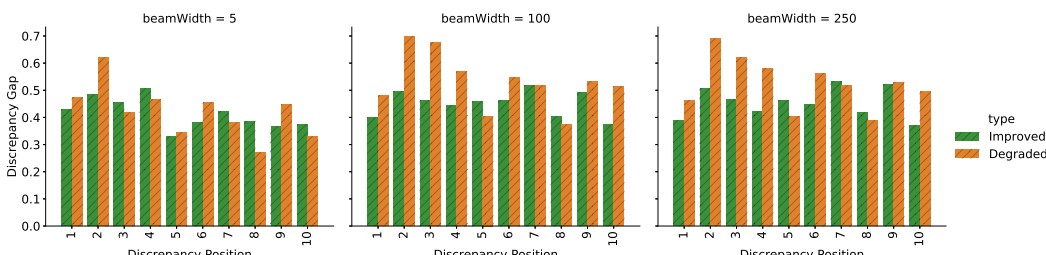

Figure 28: WMT'14 En-De: Mean discrepancy gap per position ($\mathcal{N} = 2$).

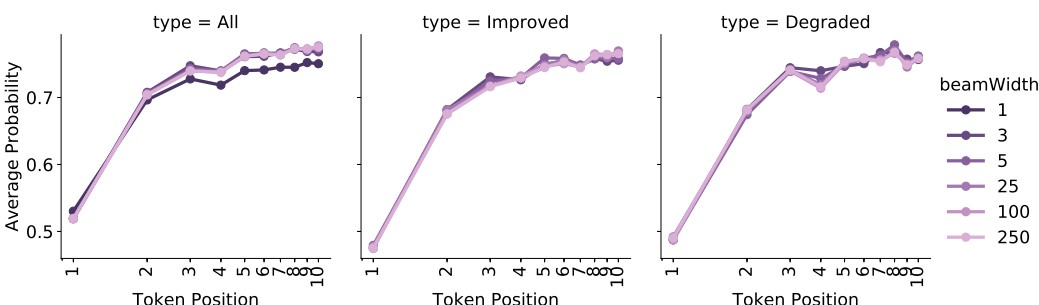

Figure 29: WMT'14 En-De: Average token probability per position ($\mathcal{N} = 2$).

## D   ANALYSIS OF LENGTH BIAS

Table 8 shows the mean length of generated sentences for different beam widths for the baseline, normalized to the best tested beam width. All values are very close 1.0, which suggest that the observed performance degradation is not due to length bias. We note that for machine translation and summarization this is due to the use of length normalization on the hypotheses log-likelihood, as suggested by Koehn & Knowles (2017) (without normalization, the performance degradation would have been worse).[4] In image captioning, however, there is no observed length bias even when length normalization is not used.[5]

In Section 4.1, we showed substantial performance degradation as we increase the beam width. As the results in Table 8 demonstrate that there is no significant change in the length of generated sequences, the observed performance degradation cannot be attributed to length bias.

Table 8: Analysis of the mean length, normalized to best test width (in bold).

| Task | Dataset | $B$=1 | $B$=3 | $B$=5 | $B$=25 | $B$=100 | $B$=250 |
|---|---|---|---|---|---|---|---|
| Translation | En-De | 0.99 | 1.0 | **1.0** | 1.0 | 0.99 | 0.98 |
| | En-Fr | 0.99 | 1.0 | **1.0** | 1.0 | 0.99 | 0.91 |
| Summarization | Gigaword | 1.03 | **1.0** | 0.99 | 0.99 | 1.0 | 1.01 |
| Captioning | MSCOCO | 1.04 | **1.0** | 0.99 | 0.98 | 0.98 | 0.98 |

## E   IMAGE CAPTIONING: CIDER AND SPICE

Table 9 compares the baseline vs. the constrained beam search methods on the MSCOCO image caption task using the metrics CIDEr and SPICE. The results show similar trends to those observed for BLEU in Section 5. In particular, we see that the performance degradation for larger beams also occurs for CIDEr and SPICE in the baseline, and that our gap constraint method eliminates this degradation. Similar to our results for BLEU, we note that our rank constraint is not as effective as our gap constraint for the image captioning task.

Table 9: Evaluation of image captioning on MSCOCO dataset using the CIDEr and SPICE metrics (higher values are better, best baseline in bold).

| Dataset | Method | Threshold | $B$=1 | $B$=3 | $B$=5 | $B$=25 | $B$=100 | $B$=250 |
|---|---|---|---|---|---|---|---|---|
| CIDEr | Baseline | | 0.974 | **1.018** | 1.005 | 0.953 | 0.946 | 0.945 |
| | Constr. Gap | $\mathcal{M}=0.4$ | 0.974 | 1.016 | 1.018 | 1.016 | 1.016 | 1.016 |
| | Constr. Rank | $\mathcal{N}=2$ | 0 | 1.022 | 1.006 | 0.978 | 0.977 | 0.977 |
| SPICE | Baseline | | 18.13 | **18.54** | 18.43 | 17.76 | 17.68 | 17.64 |
| | Constr. Gap | $\mathcal{M}=0.45$ | 18.13 | 18.41 | 18.44 | 18.43 | 18.43 | 18.43 |
| | Constr. Rank | $\mathcal{N}=2$ | 0 | 18.60 | 18.51 | 18.15 | 18.15 | 18.15 |

---

[4]This is consistent with Ott et al.'s (2018) results on performance degradation even when using length normalization.

[5]In fact, as we stated earlier, we found that length normalization reduces the overall performance.

