# OpenReview forum: "(Unconstrained) Beam Search is Sensitive to Large Search Discrepancies"
_ICLR.cc/2019/Conference_

### Official Review · AnonReviewer2 · 2018-11-01
**A thorough analysis of (and two heuristic solutions to) the failures of beam search when applied to modern neural models**

**Rating:** 7
**Confidence:** 5

**Review:**

Pros:
- The paper generalizes upon past observations by Ott et al. that NMT models might decode "copies" (of the source sentence) when using large beam widths, which results in degraded results. In particular, the present paper observes similar shortcomings in two additional tasks (summarization and captioning), where decoding with large beam widths results in "training set predictions." It's unclear if this observation is novel, but in any case the connection between these observations across NMT and summarization/captioning tasks is novel.
- The paper draws a connection between the observed degradation and "label bias", whereby prefixes with a low likelihood are selected merely because they lead to (nearly-)deterministic transitions later in decoding.
- The paper suggests two simple heuristics for mitigating the observed degradation with large beam widths, and evaluates these heuristics across three tasks. The results are convincing.
- The paper is very well written. The analysis throughout the paper is easy to follow and convincing.

Cons:
- Although the analysis is very valuable, the quantitive impact of the proposed heuristics is relatively minor.

Comments/questions:
- In Eq. 2, consider using $v$ or $w$ for the max instead of overloading $y$.
- To save space, you might compress Figure 1 into a single figure with three differently-styled bars per position that indicate the beam width (somewhat like how Figure 3 is presented). You can do this for Figure 2 as well, and these compressed figures could then be collapsed into a single row.
- In Section 5, when describing the "Discrepancy gap" constraint, you say that you "modify Eq. 3 to include the constraint", but I suspect you meant that you modify Eq. 1 to include this constraint.
- In Table 4, why didn't you tune $\mathcal{M}$ and $\mathcal{N}$ separately for each beam width?

---

> ### Author Response · Authors · 2018-11-10
> **Thanks for your review**
>
> We thank the review for the review and the comments. We will incorporate your comments and suggestions in the final version of the paper.
>
> Regarding the concern about the minor empirical improvement:
> This work is focused on studying the previously reported search phenomenon of performance degradation in beam search with large beams. We believe understanding search-related phenomena provides deeper insight into the neural decoding process and can help design better search algorithms. Our proposed variants of discrepancy-constrained beam search are meant to "fix" the performance based on our analysis, and the success in doing so validates our analysis (together with our analysis of search discrepancies on the results of the discrepancy-constrained beam search). As such, we did not expect our methods to significantly outperform the best beam width but to eliminate the effect of the beam degradation and validate our analysis of the performance degradation phenomenon.
>
> Regarding your questions:
> - You are correct about the mistake in Section 5: we do mean "modify Eq. 1" instead of "modify Eq. 3". We will fix it.
>
> - We can definitely tune a different $\mathcal{M}$ and $\mathcal{N}$ for each beam width and we expect it to provide further improvement, however we preferred to show the robustness of a relatively simple tuning in effectively eliminating the performance degradation (this is why we are also interested in the easier-to-tune rank constraint). This is, again, consistent with our motivation stated above of studying the performance degradation, explaining this phenomenon, and mitigating it.
>
> Please let us know if you have further questions. We also ask the reviewer to see the new results we report in the response to AnonReviewer3 on Chinese translation (that shows the applicability of our analysis to languages that are significantly different than English) and the success of our methods with respect to two alternative evaluation metrics in image captioning.

---

### Official Review · AnonReviewer1 · 2018-11-02
**it is not right to do analysis on test set.**

**Rating:** 5
**Confidence:** 5

**Review:**


This work does extensive experiments on three different text generation tasks and shows the relationship between wider beam degradation and more and larger early discrepancies. This is an interesting observation but the reason behind the scene are still unclear to me. A lot of the statements in the paper lack of theoretical analysis.

The proposed solutions addressing the beam discrepancies are effective, which further proves the relationship between beam size and early discrepancies. My questions/suggestions are as follows:
* It’s better to show the dataset statistics along with Fig1,3. So that readers know how much of test set have discrepancies in early steps.
* It is not right to conduct your analysis on the test set. You have to be very clear about which results are from test set or dev set.
* All the results with BLEU score must include the brevity penalty as well. It is very useful to analyze the length ratio changes between baseline, other methods, and your proposal.
* The example in Sec. 4.6 is unclear to me, maybe you could illustrate it more clearly.
* Your approaches eliminate the discrepancies along with the diversity with a wider beam. I am curious what if you only apply those constraints on early steps.
* I suggest comparing your proposal to the word reward model in [1] since it is also about improving beam search quality. Your threshold-based method is also kind of word reward method.
* In eq.2, what do you mean by sequence y \in V? y is a sequence, V just a set of vocabulary.  What do you mean by P (y|x;{y_0..y_t}). Why the whole sequence y is conditioned on a prefix of y?

[1] Huang et al, "When to Finish? Optimal Beam Search for Neural Text Generation" 2017

---

> ### Author Response · Authors · 2018-11-10
> **Our algorithms are tuned on a held-out dev set (part 1)**
>
> Thank you for your review. We will incorporate your comments and suggestions in the final version of the paper.
>
> We would like to address the reviewer questions and concerns:
>
> - The use of test set in our analysis:
> While the empirical results in Section 4 are on the test set, our algorithms (discrepancy-constrained beam search variants) are tuned only on a dev set (with *no* information from the test set analysis). The reason why we present the empirical analysis in Section 4 on the test set is that we are focused on explaining the performance degradation that was previously reported on the test set. This analysis only provides a better understanding of the what is going on: no information from that analysis is later used in our algorithmic improvements. We did perform a similar analysis on the dev set and observed similar trends, however we thought presenting the test set is consistent with the previously reported results.
>    In Section 5, we propose two discrepancy-constrained variants of beam search. In order to tune these $\mathcal{M}$ and $\mathcal{N}$ that control the constraints, we used a held-out validation set, and then evaluated the performance on the test set (we also repeat our analysis of search discrepancy and it can be compared to the one in Section 4). Again, we would like to stress that no information from the test set were used to tune the algorithms (if the phenomenon did not occur on the dev set, the tuning would not yield useful values that will eliminate the performance degradation in the test set). We agree with the reviewer that this was not completely clear in the paper and we will clarify it in our final version.
>
> - Length ratio changes:
> We specifically addressed the topic of length ratio changes in Appendix D, to demonstrate that the performance degradation on the baseline is not due to significant bias in length. Since the reviewer asked about the length ratio changes in our algorithms as well, we provide an updated table that also includes our discrepancy-constrained algorithms. Note that this is not the brevity term but it shows whether the length of the prediction has changed between the different beam widths and when comparing the baseline to our algorithms (and allows us to analyze other metrics that are not BLEU).
>    The table below includes the average length relative to the average length of the top performing baseline configuration (marked with *). As we state in Appendix D, the performance degradation in the baseline is not due to significant change in the length ratio. We can also see that our algorithm keeps nearly the same length ratio across the different beam widths (and is even marginally more stable than the baseline).
>
>
> +----------+--------------+------+-------+-------+------+-------+-------+
> | Dataset  |  Algorithm   | B=1  |  B=3  |  B=5  | B=25 | B=100 | B=250 |
> +----------+--------------+------+-------+-------+------+-------+-------+
> | En-De    | Baseline     | 0.99 | 1.00  | 1.00* | 1.00 |  0.99 |  0.98 |
> | En-De    | Constr. Gap  | 0.99 | 1.00  | 1.00  | 1.00 |  1.00 |  1.00 |
> | En-De    | Constr. Rank | 0.99 | 1.00  | 1.00  | 1.00 |  1.00 |  0.99 |
> +----------+--------------+------+-------+-------+------+-------+-------+
> | En-Fr    | Baseline     | 0.99 | 1.00  | 1.00* | 1.00 |  0.99 |  0.91 |
> | En-Fr    | Constr. Gap  | 0.99 | 1.00  | 1.00  | 1.00 |  1.00 |  0.99 |
> | En-Fr    | Constr. Rank | 0.99 | 1.00  | 1.00  | 1.00 |  1.00 |  1.00 |
> +----------+--------------+------+-------+-------+------+-------+-------+
> | Gigaword | Baseline     | 1.03 | 1.00* | 0.99  | 0.99 |  1.00 |  1.01 |
> | Gigaword | Constr. Gap  | 1.03 | 0.99  | 0.99  | 0.99 |  0.99 |  0.99 |
> | Gigaword | Constr. Rank | 1.03 | 1.00  | 0.99  | 0.99 |  0.99 |  0.99 |
> +----------+--------------+------+-------+-------+------+-------+-------+
> | COCO     | Baseline     | 1.04 | 1.00* | 0.99  | 0.98 |  0.98 |  0.98 |
> | COCO     | Constr. Gap  | 1.04 | 1.00  | 0.99  | 0.99 |  0.99 |  0.99 |
> | COCO     | Constr. Rank | 1.04 | 1.00  | 0.99  | 0.99 |  0.99 |  0.99 |
> +----------+--------------+------+-------+-------+------+-------+-------+

---

> ### Author Response · Authors · 2018-11-10
> **Our algorithms are tuned on a held-out dev set (part 2)**
>
> (This is part 2 of our response)
>
> - Example in Section 4.6:
> In Section 4.6 we consider one case of training set prediction and explain how it exhibits our analysis on the search discrepancies. For the Gigaword dataset, we found that increasing the beam width leads to more predictions of "<weekday>'s sports scoreboard" that are all evaluated poorly against their reference (note that for lower beam widths we do not observe such predictions at all).
>    Our analysis in Section 4 explains that increasing the beam width can lead to large search discrepancies that are being selected since they are followed by a sequence of high (conditional) probability tokens that yield overall higher probability.
>     In Section 4.6 we present evidence that this scenario explains the increasing frequency of this (incorrect) summary:
> 	* The first token, <weekday>, has a relatively low probability (an average a discrepancy gap of 3.63 vs. a first token discrepancy gap average of 0.39 across the dataset).
> 	* In order for a sequence with such a low-probability first token to be chosen the most likely, the rest of tokens are significantly higher. The reason is that the training set has 2971 instances of "<weekday>'s sports scoreboard" as a target. This makes the "sports" and then "scoreboard" very likely, conditioned on the prefix. Due to this exposure bias in the training set and the fact that probabilities are locally normalized to 1 (label bias), these consecutive tokens of the low-probability first one end up "contributing" much more to the overall probability compared to consecutive tokens for the higher probability first token, leading this (incorrect) summary to be the most likely one.
> 	* For lower beam width, this first low-probability token would not have been considered as it would not be one of the top B.
>
>
> - Applying the constraints on early steps only:
> Thanks for the interesting question. We have performed an initial analysis of the result when only constraining the early positions. While it is still beneficial to do it, it seems that it might not be enough to mitigate the performance degradation as new discrepancies will appear in the positions following the ones that are constrained (can be thought of "early" positions after the constrained ones).
>     A thorough analysis requires more time. We will do the analysis by the end of the review period and if accepted, we will include it as an appendix in the final version of the paper.
>
>
> - Comparison to Huang et al.:
> Our work is focused on the analyzing and eliminating the problem of performance degradation in large beam width. The problem addressed by Huang et al. is when to stop searching for new hypotheses when we already have completed hypotheses. Specifically, their work does not deal with large beam widths (up to 20 in their work) and as a result they do not observe a performance degradation. Trying to implement our constraints into other beam search algorithms (such as Huang et al.'s) is a direction for future work.
>
>
> - Clarifying eq.2:
> The $y$ inside is meant to be a token rather than a sequence (which we denote $\ry$; we mistakenly refer to it as $y$ in the line before the equation), and the equation simply means that the conditional probability of token y_t in the generated sequence is smaller than the conditional probability of the token with the highest conditional probability. We understand that the current notation is a bit confusing and we will revise it to make it clear.
>
>
> Please let us know if you have further questions. We also ask the reviewer to see the new results we report in the response to AnonReviewer3 on Chinese translation (that shows the applicability of our analysis to languages that are significantly different from English) and the success of our methods with respect to two alternative evaluation metrics in image captioning.

---

### Official Review · AnonReviewer3 · 2018-11-03
**Interesting direction, although more work required**

**Rating:** 5
**Confidence:** 4

**Review:**

This paper addresses issues with the beam search decoding algorithm that is commonly applied to recurrent models during inference. In particular, the paper investigates why using larger beam widths, resulting in output sequences with higher log-probabilities, often leads to worse performance on evaluation metrics of interest such as BLEU. The paper argues that this effect is related to ‘search discrepancies’ (deviations from greedy choices early in decoding), and proposes a constrained decoding mechanism as a heuristic fix.

Strengths:
- The reduction in performance from using larger beam widths has been often reported and needs more investigation.
- The paper views beam search decoding through the lens of heuristic and combinatorial search, and suggests an interesting connection with methods such as limited discrepancy search (Harvey and Ginsberg 1995) that seek to eliminate early ‘wrong turns’.
- In most areas the paper is clear and well-written, although it may help to be more careful about explaining and / or defining terms such as ‘highly non-greedy’, ‘search discrepancies’ in the introduction.

Weaknesses and suggestions for improvement:

- Understanding: The paper does not offer much in the way of a deeper understanding of search discrepancies. For example, are search discrepancies caused by exposure bias or label bias, i.e. an artifact of local normalization at each time step during training, as suggested in the conclusion? Or are they actually a linguistic phenomenon (noting that English, French and German have common roots)? As there are neural network methods that attempt to do approximate global normalization (e.g. https://www.aclweb.org/anthology/P16-1231), there may be ways to investigate this question by looking at whether search discrepancies are reduced in these models (although I haven’t looked deeply into this).

- Evaluation: In the empirical evaluation, the results seem quite marginal. Taking the best performing beam size for the proposed method, and comparing the score to the best performing beam size for the baseline, the scores appear to be within around 1% for each task. Although the proposed method allows larger beam widths to be used without degradation during decoding, of course this is not actually beneficial unless the larger beam can improve the score. In the end, the evidence that search discrepancies are the cause of the problems with large beam widths, and therefore the best way to mitigate these problems, is not that strong.

- Evaluation metrics and need for human evals: The limitations of automatic linguistic evaluations such as BLEU are well known. For image captioning, the SPICE (ECCV 2016 https://arxiv.org/abs/1607.08822) and CIDEr (CVPR 2015 https://arxiv.org/abs/1411.5726) metrics show much greater correlation with human judgements of caption quality, and should be reported in preference (or in addition) to BLEU. More generally, it is quite possible that the proposed fix based on constraining discrepancies could improve the generated output in the eyes of humans, even if this is not strongly reflected in automatic evaluation metrics. Therefore, it would be interesting to see human evaluations for the generated outputs in each task.

- Rare words: The authors reference Koehn and Knowles’ (2017) six challenges for NMT, which includes beam search decoding. One of the other six challenges is low-frequency words. However, the impact of the proposed constrained decoding approach on the generation of rare words is not explored. It seems reasonable that limiting search discrepancies might also further limit the generation of rare words. Therefore, I would like to suggest that an analysis of the diversity of the generated outputs for each approach be included in the evaluation.

- Constrained beam search: There is a bunch of prior work on constrained beam search. For example, an algorithm called constrained beam search was introduced at EMNLP 2017 (http://aclweb.org/anthology/D17-1098). This is a general algorithm for decoding RNNs with constraints defined by a finite state acceptor. Other works have also been proposed that are variations on this idea, e.g. http://aclweb.org/anthology/P17-1141, http://aclweb.org/anthology/N18-1119). It might be helpful to identify these in the related work section to help limit confusion when talking about this ‘constrained beam search’ algorithm.

Minor issues:
- Section 3. The image captioning splits used by Xu et al. 2015 were actually first proposed by Karpathy & Li, ‘Deep visual-semantic alignments for generating image descriptions’, CVPR 2015, and should be cited as such. (Some papers actually refer to them as the ‘Karpathy splits’.)
- In Table 4 it is somewhat difficult to interpret the comparison between the baseline results and the constrained beam search methods, because the best results appear in different columns. Bolding the highest score in every row would be helpful.

Summary:
In summary, improving beam search is an important direction, and to the best of my knowledge the idea of looking at beam search through the lens of search discrepancies is novel. Having said, I don't feel that this paper in it's current form contributes very much to our understanding of RNN decoding, since it is not clear if search discrepancies are actually a problem. Limiting search discrepancies during decoding has minimal impact on BLEU scores, and it seems possible that search discrepancies could just be an aspect of linguistic structure. I rate this paper marginally below acceptance, although I would encourage the authors to keep working in this direction and have tried to provide some suggestions for improvement.

---

> ### Author Response · Authors · 2018-11-10
> **Addressing the reviewer concerns**
>
> Thank you for your detailed review. We will incorporate your comments and suggestions in the final version of the paper.
>
> We would like to address the reviewer concerns:
>
> - Understanding the discrepancy phenomenon and applicability to languages significantly different from English:
> We believe that the search discrepancies occur due to the combination of exposure and label bias as explained in our discussion. We provide a concrete example in Section 4.6 (see our response for AnonReviewer 1 for a detailed explanation of this example).
>     The reviewer raises a potential concern that the discrepancy phenomenon is actually a linguistic phenomenon associated with English or similar languages. We therefore perform an experiment of generating translations in a language that is significantly different than English: Chinese. We train and evaluate the convolutional translation model by Gehring et al. (2017) on the WMT'17 En-Zh dataset. We performed the analysis in the paper on this dataset and found similar trends (results follow). Specifically:
> 	* The dataset exhibits significant performance degradation for large beam width
> 	* Our analysis of the frequency and size of the early discrepancies and the comparison between improved vs. degraded sequences yielded similar trends to the other languages
> 	* We used a gap-constrained beam search, tuned on a held-out validation set, and successfully eliminated the performance degradation. We will have results on the rank constraint soon.
>
> The results for the baseline vs. the discrepancy-constrained beam search are described in the following table:
>
> |    Dataset     |   Method    |    Threshold    |  B=1  |  B=3  |  B=5  | B=25  | B=100 | B=250 |
>
> | En-Zh (BLEU-4) | Baseline    |                 | 15.20 | 17.47 | 17.68 | 16.48 |  9.44 |  6.28 |
> | En-Zh (BLEU-4) | Constr. Gap | \mathcal{M}=1.0 | 15.20 | 15.49 | 17.71 | 17.73 | 17.79 | 17.83 |
>
>
> - Evaluation metrics and need for human evals:
> Our work is done from a search perspective and is focused on analyzing, explaining, and eliminating the performance degradation in beam search with respect to a given evaluation metric. Previous work reporting the performance degradation are also based only on automatic evaluation.
>     The reviewer raises a potential concern regarding using BLEU-4 to evaluate image captioning, as there are indication that CIDEr and SPICE are better correlated with human judgment. We therefore present results for these two metrics on the image captioning dataset (COCO). Our analysis finds similar results to the ones found for BLEU-4:
> 	* There is a significant performance degradation for the image captioning tasks, with respect to both CIDEr and SPICE
> 	* We used a gap-constrained beam search, tuned on a held-out validation set, and significantly reduced (and almost eliminated) the performance degradation.
>
> The results for the baseline vs. the discrepancy-constrained beam search are described in the following table:
>
> |   Dataset    |   Method    |    Threshold     |  B=1  |  B=3  |  B=5  | B=25  | B=100 | B=250 |
>
> | COCO (CIDEr) | Baseline    |                  | 0.974 | 1.018 | 1.005 | 0.953 | 0.946 | 0.945 |
> | COCO (CIDEr) | Constr. Gap | \mathcal{M}=0.4  | 0.974 | 1.016 | 1.018 | 1.016 | 1.016 | 1.016 |
> | COCO (SPICE) | Baseline    |                  | 18.13 | 18.54 | 18.43 | 17.76 | 17.68 | 17.64 |
> | COCO (SPICE) | Constr. Gap | \mathcal{M}=0.45 | 18.13 | 18.41 | 18.44 | 18.43 | 18.43 | 18.43 |
>
> - Rare words:
> 	* One main problem with rare words is the limited size of vocabulary (Koehn and Knowles, 2017). We do not change the vocabulary size. In fact, our machine translation model is using BPE (Sennrich et al., 2016) to allow translation of rare words using subword units.
> 	* Regarding the frequency of rare words that are in the vocabulary, we are definitely reducing the frequency of such rare words compared to the large beams of the baseline: for example, "copies" are all sequences of rare words that we eliminate. However, we analyzed the occurrence of rare words that are also in the reference for En-De translation and found no significant difference between the baseline and our algorithm in that respect. Note that the baseline itself poorly represents rare words that are indeed in the reference.
> 	* For image captioning and summarization, the related problem of novel captions/summaries vs. ones from the training set is thoroughly discussed in the paper (see Section 4.5 and Appendix B).
>
> - Constrained beam search:
> Thanks for pointing that out. We are familiar with these works but did not include them as they are not directly related (they are addressing different problems such as forcing the inclusion of a selected token in the sequence). However, we agree with the reviewer that mentioning these works in the related work section will help reduce confusion with other constrained variants of beam search. We will add these to the related work section.
>
> Please let us know if you have further questions.

---

> > ### Comment · AnonReviewer3 · 2018-12-11
> > **Thank you for your comprehensive response to my comments**
> >
> >
> > The additional results for English:Chinese, and additional metrics, references etc. largely resolve some of my smaller concerns. I guess my remaining (more significant concern) is about the effectiveness of the approach. Some techniques are important because they are principled and well-motivated. Other techniques might be heuristics, but they can still be very important if they are effective. In this case, the proposed approach is a heuristic, but it is not particularly effective at improving the scores of the resulting sequences. In my review, I argued that 'Although the proposed method allows larger beam widths to be used without degradation during decoding, of course this is not actually beneficial unless the larger beam can improve the score.' I would still maintain this view, and I would disagree with the the authors comment to the AC (above) that 'Our search algorithms are not supposed to find solutions with higher search score', since I cannot see having higher evaluation score for larger beams is beneficial, unless those scores are also the highest overall (across all lengths).
> >
> > In summary, I will retain my rating - marginally below acceptance.

---

> > > ### Author Response · Authors · 2018-12-12
> > > **Thanks for your comment**
> > >
> > > Thanks for your comment, we are glad that we have largely resolved some of your concerns.
> > >
> > > For your remaining concern, we think the dichotomy you propose ("principled and well-motivated" vs. "heuristic") is a false one. Our approach is both. Through extensive experiments, we've provided an understanding of one of the "six challenges for neural machine translation" (Koehn & Knowles, 2017), shown that it is a much more general phenomenon than the existence of "copies" (the current best explanation), and shown that the same understanding applies to two very different sequence-to-sequence neural decoding tasks (summarization and captioning). We think we've established the novel principles and motivations for our algorithmic approach.
> > >
> > > Our heuristic approach demonstrates how the degradation of wider beam widths can be removed but does not unambiguously lead to higher evaluation. While we would have obviously liked our approach to do so, the more important contribution is the analysis and identification of this search phenomenon which is (apparently) widespread in neural sequence decoding tasks. Other researchers (and we too) can now work on the development of better algorithmic approaches informed by the deeper understanding.
> > >
> > > We believe that the analysis and heuristic approach together represent a contribution significant enough to merit acceptance.

---

### Comment · Area_Chair1 · 2018-11-09
**Distinction between search errors and modeling errors not clear**

This paper investigates the problem of search degrading translation accuracy as measured by BLEU score.

However, the paper seems to be conflating two fundamental issues, search errors and modeling errors. Search errors are errors where the search algorithm is not able to find the highest-scoring hypothesis, and modeling errors are errors where the highest-scoring hypothesis is actually not a good one according to the model.

The widely-known problem of BLEU (or other) scores degrading with larger beams is due to modeling errors: MLE-trained models tend to prefer shorter sentences, and this can be (largely) fixed by better modeling of length. The simplest method for doing so is length normalization, searching for the hypothesis that has the highest average likelihood per word rather than the highest likelihood overall per sentence (see "On the Properties of Neural Machine Translation: Encoder–Decoder Approaches" SSST 2014). This largely fixes the problem of large beams degrading accuracy (see "Six Challenges for Neural Machine Translation" WNMT 2017), although there are other methods for length normalization as well.

In contrast, this paper attempts to indirectly fix the problem of modeling errors by changing the search algorithm. Hobbling the search algorithm seems like a rather indirect way to solve a problem that is essentially a modeling problem (and has already been largely fixed by other methods). In addition, the discussion seems pretty incomplete without a discussion of whether the search algorithm is actually achieving better model scores, which is the fundamental job of the search algorithm in the first place.

It would be nice to see a discussion of these issues in the author response if possible.

---

> ### Author Response · Authors · 2018-11-10
> **Length Bias and Search vs. Model errors**
>
> Thanks for your comments. We will address the two issues separately.
>
> Length bias
> The review claims that the problem of performance degradation is well known and is due to the fact that MLE-trained models tend to prefer shorter sentences. However, this is not the case we are addressing. We are considering the case of performance degradation that is *not* due to length bias. Even when length-normalization is used, performance degradation is observed in previous work that we cite and in our baseline results (which explicitly use length-normalization). More specifically:
> 	* For the translation and summarization models we perform length normalization as a baseline (see Section 3) and observe search degradation.
> 	* Appendix D analyzes the length bias and shows that the performance degradation we observe is not associated with significant length bias.
> 	* The problem of ``copies'' (that we generalize in this work) was observed by Ott et al. (2018) when using length normalization.
> 	* In "Six Challenges for Neural Machine Translation" (Koehn & Knowles, 2017) that the reviewer refers to, the authors show that while normalization reduces the problem of performance degradation of beam search, it does not eliminate it.  As a consequebnce, performance degradation is still listed as Challenge #6.
> 	* Besides BLEU we analyze rouge for summarization, and in our response for AnonReviewer3 we added CIDEr and SPICE for image captioning.
>
>
> Model errors vs. search errors
> We agree with the distinction between search errors and modeling errors and think this distinction is useful in highlighting our contributions. We do not agree that we are conflating model error and search error.
>    	 In the context of beam search, wider beams lead to higher likelihood hypotheses (fairly trivially because we are searching a larger space). The phenomenon we are exploring is the observation over a number of tasks that the higher likelihood hypotheses result in lower quality results due to the misalignment between the learned probability model and metric; that is, due to model errors.
> 	Our position is that modeling errors are unavoidable (due to e.g., noisy training data, training data that is necessarily a small sample of a huge space, etc.).  By understanding how these modeling errors interact with search we can improve task performance. Note that we do not consider search discrepancies as ``search errors''. The notion of discrepancies is well established in the search literature and limited discrepancy search is meant to deal with ``heuristic mistakes'': decision points in a tree search where the guiding heuristic prefers what is not actually the best option in terms of the final solution quality. We believe that there is a meaningful analogy here: just as a human-designed heuristic is not infallible (e.g. due to a myopic perspective), the learned model is not infallible due to modeling errors.
> 	We are proposing a more nuanced search, based on our analysis and identification of a common phenomenon, to better deal with the modeling error. We note that the previous works on "copies" and training set predictions also addressed these problems using changes in search. Ott et al. (2018) added a pruning constraint to the beam search, while Vinyals et al. (2017) intentionally reduce the beam size to avoid training set predictions. However, these changes were aimed at the narrow phenomena observed (copies and training set predictions) which we generalize.
> 	Section 6 discusses the cause for the observed phenomenon, namely a combination of exposure bias and label bias. Our example in Section 4.6 (that is further clarified in our response to AnonReviewer 1) provide a detailed example.

---

> > ### Comment · Area_Chair1 · 2018-11-19
> > **To further disentangle the effects**
> >
> > First, thank you for the response and clarification regarding length normalization. I had missed the detail in the paper.
> >
> > Second, as a suggestion to disentangle the effect of search and model errors, I have a simple suggestion: could you please report the value of the search criterion (in this case, "length normalized model score") for each of the search algorithms? This would help show whether the algorithms you're proposing are actually better search algorithms, or whether they're exploiting some systematic difference between model scores and outputs that give high BLEU scores.
> >
> > Third, while I focused on length normalization in my previous comment, this is actually a band-aid over the true problem of not optimizing directly for the evaluation score. There are more direct methods to do so (e.g. "sequence level training for recurrent neural networks" by Ranzato et al.), and this made me wonder whether the proposed algorithm would be useful in a situation where the model has been tuned with one of these objectives.

---

> > > ### Author Response · Authors · 2018-11-21
> > > **Thanks for your questions**
> > >
> > > Thanks for your questions.
> > >
> > > Regarding disentangling the effects:
> > > Our algorithms do not find solutions with higher search score (i.e., normalized log-prob). Intuitively, we are constraining the explored hypotheses space and it can lead to decreased search score. However, due to modeling errors discussed in our previous response, they do lead to a higher evaluation (BLEU, Rouge, SPICE or CIDEr) for larger beams. Section 4.4 explains how degraded solutions (i.e., solutions with lower evaluation score) end up getting a higher search score and, together with the example in Section 4.6, provides the intuition for why we propose to constrain the hypotheses space and effectively prefer, in some cases, lower search score solutions. Our search algorithms are not supposed to find solutions with higher search score. They are supposed to mitigate the effect of the large search discrepancies, lead to a higher evaluation score for larger beams, and validate our empirical analysis in Section 4.
> > >
> > >
> > > Regarding methods like Ranzato et al. (2016):
> > > This work is focused on studying the previously reported search phenomenon of performance degradation in beam search with large beams. We believe understanding search-related phenomena provides deeper insight into the neural decoding process and can help design better search algorithms (see our response to AnonReviewer2 for a detailed motivation). As Ott et al. (2018) pointed out, Ranzato et al. (2016) do not provide an analysis of the problem or the impact of their solution on the search space. Specifically, as they only analyze beam search up to a width of 10, we cannot tell if performance degradation occurs and if our methods are needed. However, we believe the framework of search discrepancies will be useful in analyzing search-based neural decoding algorithms beyond the phenomenon of the performance degradation phenomenon - but this remains to be demonstrated in the future. Similarly, applying a similar analysis to other works such as Ranzato et al. (2016) is a direction for future work.

---

### Meta-Review · Area_Chair1 · 2018-12-13
**Interesting insights but heuristics in approach worrisome**

**Confidence:** 4
**Recommendation:** Reject

**Metareview:**

This paper examines a concept (also coined by the paper) of "search discrepancies" where the search algorithm behaves differently with large beam sizes. It then proposes heuristics to help prevent the model from performing worse when the size of the beam is increased.

I think there are some interesting insights in this paper with respect to how search works in modern neural models, but most reviewers (and me) were concerned by the heuristic approach taken to fix these errors. I still think that within a search paper, a clear separation between modeling errors and search errors is useful, and adding heuristics on top has a potential to making things more complicated down the road when, for example, we change our model or we change our training algorithm.

It would be nice if the nice insights in the paper could be turned into a more theoretically clean framework that could be re-submitted to a future conference.